# Comparative Transcriptome Analysis Reveals Changes in Gene Expression Associated with Anthocyanin Metabolism in *Begonia semperflorens* under Light Conditions

**Kunkun Zhao** [1], **Airong Liu** [2,*], **Yuanbing Zhang** [1], **Weixin Liu** [3] , **Zhimin Zhao** [2] **and Shuyue Yang** [2]

[1] College of Architecture, Anhui Science and Technology University, Donghua Road No. 9, Chuzhou 233100, China; kunkunzhao2021@163.com (K.Z.); zyb2246@163.com (Y.Z.)

[2] College of Life and Health Sciences, Anhui Science and Technology University, Donghua Road No. 9, Chuzhou 233100, China; 19556958506@163.com (Z.Z.); 19565985302@163.com (S.Y.)

[3] Zhejiang Provincial Key Laboratory of Tree Breeding, Research Institute of Subtropical Forestry of Chinese Academy of Forestry, Fuyang 311400, China; lwx060624@163.com

[*] Correspondence: liuar@ahstu.edu.cn; Tel.: +86-(0552)-3197206

**Abstract:** Anthocyanins, recognized as stress indicators, particularly under high-light conditions, play a pivotal role in plant stress responses. The advent of transcriptomics has opened avenues to elucidate the mechanisms underlying high light-induced anthocyanin biosynthesis. This study delved into transcriptomic changes in *Begonia semperflorens* leaves under varying light intensities: 950–9600 lx (TL_100), 6800–7000 lx (HS_75), and 4300–4500 lx (LS_25). To confirm the expression profiles of the key genes, we chose 12 critical genes associated with anthocyanin production for quantitative reverse transcription PCR (qRT-qPCR) analysis. Following this, we measured the levels of anthocyanins to substantiate the findings from the gene expression analysis. The transcriptome assembly in this study was extensive, yielding 43,038 unigenes that collectively spanned about 49.83 million base pairs, with an average unigene length of 1157 bp and an N50 value of 1685 bp. This assembly facilitated a thorough functional annotation across seven distinct protein databases, leading to the classification of 16,363 unigenes into 58 different families of transcription factors. Our comparative analysis of the transcriptomes highlighted a substantial number of differentially expressed genes (DEGs): 5411 DEGs between HS_75 and TL_100 conditions, with 3078 showing increased expression and 2333 showing decreased expression; 4701 DEGs between LS_25 and TL_100, consisting of 2648 up-regulated and 2053 down-regulated genes; and 6558 DEGs between LS_25 and HS_75, with 3032 genes up-regulated and 3526 down-regulated. These DEGs were significantly involved in critical pathways, such as anthocyanin synthesis, plant hormone signaling, and other regulatory mechanisms. This study suggests that genes, including *F3′H*, *MYB102*, and *SWEET1*, could play vital roles in regulating anthocyanin synthesis in response to various light conditions, potentially impacting the expression levels of other genes, like *WRKYs*, *ATHB12*, and those similar to *HSP*.

**Keywords:** ornamental plants; transcriptomics; abiotic stress; high light exposure; leaf color

## 1. Introduction

Ornamental plants, renowned for their diverse colors and patterns, significantly contribute to the aesthetics and commercial success within the floriculture industry [1,2]. Predominant pigments, like chlorophyll, flavonoids, carotenoids, and betalains, define plant coloration, each stemming from distinct metabolic pathways [3,4]. Flavonoids, a prevalent class of secondary metabolites in angiosperms, exhibit a wide spectrum of colors [5]. These compounds are crucial for various physiological roles in plants, including pigmentation, resilience against biotic and abiotic stressors, and developmental regulation. For instance, studies have shown that flavonoids in *Ginkgo biloba* confer resistance to low temperatures and salt stress by scavenging free radicals [6] and offer protection

against insect predation [7]. In *Oryza sativa*, the flavonoids present in phloem or leaves can deter pests like *Niloparvata lugens* or *Ditylenchus angustus*, inhibit the growth of fungi and bacteria, and counteract other pathogenic organisms [8,9]. In *Arabidopsis*, flavonols are known to influence auxin transport, impacting root growth, anther dehiscence, and pollen germination [10].

Anthocyanins, a subset of flavonoids, are water-soluble pigments responsible for vibrant hues, ranging from orange to blue in leaves, flowers, and fruits [4,11]. These compounds serve as vital secondary metabolites, imparting color to plant tissues and protecting against the oxidative stress caused by environmental factors, such as salinity, low temperature, drought, and high light [12]. They also safeguard against ultraviolet radiation and help prevent oxidative damage to DNA [7,13]. Furthermore, anthocyanins in *Arabidopsis* seeds play a role in regulating dormancy and germination [14]. Beyond their aesthetic value in ornamental plants, flavonoids have been recognized for their antioxidant properties, which contribute to neuroprotection, anti-inflammatory effects, pain relief, antibacterial and antispasmodic activities, cancer cell growth inhibition, and cardiovascular disease prevention [8,15].

*B. semperflorens*, a perennial evergreen herb belonging to the Begonia family, is extensively used for ornamental purposes in gardens and lawns. Its leaves often exhibit red pigmentation, primarily due to the accumulation of anthocyanins, in response to environmental triggers, such as low temperature, intense light, malnutrition, and physical injury. This makes *B. semperflorens* an excellent subject for investigating environmentally induced anthocyanin synthesis. However, detailed genetic information on many Begoniaceae species remains scarce. Transcriptome sequencing has emerged as an accurate and reliable method for exploring genomic characteristics, particularly for plants with unsequenced genomes, aiding in the comprehensive understanding of plant growth and regulatory mechanisms [16,17]. Under a low-nitrogen environment, the *PAL* (*phenylalanine ammonia-lyase*), *CHS* (*chalcone synthase*), and *DFR* (*dihydroflavonol 4-reductase*) genes are responsible for accumulating flavonoids in snow chrysanthemum (*Coreopsis tinctoria* Nutt.); this was revealed through both flavonoid metabolism and transcriptomics [16]. In *Chrysanthemum morifolium*, a combination of genomic and transcriptomic approaches demonstrates that the *C. morifolium* genome can be used to identify genes underlying key ornamental traits [17].

Previous research has demonstrated that lower temperatures increase the anthocyanin content in *B. semperflorens* seedlings, with reactive oxygen species (ROS) produced by BsR-BOHD playing a crucial role in this process [18,19]. Additionally, short day conditions have been identified as essential for anthocyanin biosynthesis in Begonia seedlings under low temperatures [19]. When the shading degree was increased, the flower number, chlorophyll content, peroxidase activity, and nitrate reductase began to increase and then decreased, while the anthocyanin content decreased gradually [20]. However, the specific mechanisms through which light influences anthocyanin biosynthesis remain under-explored. In this study, we analyzed the transcriptomes of adult *B. semperflorens* leaves under various light conditions—TL_100 (9500–9600 lx), HS_75 (6800–7000 lx), and LS_25 (4300–4500 lx)—to unravel the biological mechanisms governing anthocyanin biosynthesis in response to light.

## 2. Methods

### 2.1. Plant Material and Treatment

*B. semperflorens* plants were grown in plastic containers filled with a 7:3 mixture of peat and vermiculite, with three plants per pot. These were then acclimatized to natural environmental conditions. After 20 d, the plants underwent different light treatments using shading nets set 1.5 m above the ground, creating three light intensity environments: TL_100 (9500–9600 lx), HS_75 (6800–7000 lx), and LS_25 (4300–4500 lx). Each treatment group consisted of twenty pots. Then, 30 d later, the third and fourth leaves from each plant were collected, instantly frozen in liquid nitrogen, and stored at −80 °C for RNA extraction. For RNA-seq analysis, three biological replicates were used per treatment group.

### 2.2. Sequencing and De Novo Splicing

The total RNA was isolated and contaminating DNA was removed using DNase. Eukaryotic mRNAs were then enriched using Oligo (DT) magnetic beads. Following this, single-stranded cDNA was synthesized using random primers, which was then converted into double-stranded cDNA. The double-stranded cDNA underwent end repair, a-tailing, adapter ligation, size selection, and PCR amplification. Sequencing was carried out using the Illumina HiSeq X Ten platform, generating 150 bp paired-end reads. Unigenes were identified using Trinity software (version 2.4.0) for splice assembly, selecting the longest sequence per gene based on similarity and length [21]. CD-HIT software (version 4.6) was employed to cluster sequences and eliminate redundancy [22].

### 2.3. Database Annotation and Unigene Expression Analysis

Unigenes were annotated using various databases, including NR, KOG, GO, Swiss-Prot, eggNOG, KEGG, and the Pfam database, employing diamond (version 0.4.7) and HMMER software (version 3.0) [23,24]. The abundance of each unigene's expression was quantified using sequence similarity comparison methods [25]. Bowtie2 software (version 2.3.3.1) was used to map reads to unigenes and expression levels were determined by calculating Fragments Per Kilobase of transcript per million mapped reads (FPKMs) using eXpress software (version 1.5.1) [26,27]. DESeq software (version 1.18.0) was used to normalize unigene counts, calculate expression multiples, and assess the significance of read number differences using a negative binomial distribution [28]. Significantly different expression was defined by a *p*-value < 0.05 and an absolute log2 fold change >2.

### 2.4. Quantitative RT-PCR

For qRT-PCR analysis, we collected the third leaf from each *B. semperflorens* plant. Tissue samples, comprising a combination of 3 to 5 plants, represented a single biological replicate. And 1 microgram of RNA was used for cDNA synthesis via reverse transcription with PrimeScript®Reverse Transcriptase. Primer design was facilitated by Primer 5.0 software. Each tissue sample was subjected to qRT-PCR in triplicate, and the reaction conditions were as follows: 95 °C/10 s, 40 cycles of 95 °C/10 s, and 60 °C/45 s. Relative abundance of transcripts was determined using the $2^{-\Delta\Delta Ct}$ method [29], with the *B. semperflorens* 18S rRNA gene serving as the reference.

### 2.5. Anthocyanin Quantification

We quantified anthocyanin content following the protocol by Mita et al. [30]. Leaf tissue, weighing 0.3 g, was homogenized in 3 mL of 1% HCl in methanol and incubated in darkness at 4 °C for 24 h. After centrifugation at $3500\times g$ for 15 min, we measured the absorbance of the supernatant at 530 nm and 657 nm. Anthocyanin content was calculated as the absorbance unit (A530–0.25 × A657) per mL of the extraction solution.

## 3. Results

### 3.1. Anthocyanin Quantification, RNA-Seq, and Sequence Assembly

Exposure to light induced a red pigmentation in *B. semperflorens* leaves, with the TL_100 treatment resulting in entirely red leaves (Figure 1A). Correspondingly, the anthocyanin content in the leaves increased with light intensity (Figure 1B). Leaf samples from the three light treatments (LS_25, HS_75, and TL_100), including biological replicates, were used to construct the libraries using Illumina sequencing.

In this study, a total of nine samples (SJHT01–SJHT09) were analyzed (Tables 1 and S1). Trimmomatic software (version 0.36) was employed to preprocess the quality of the raw data [31]. Low-quality bases at the 3′ and 5′ ends were trimmed and the number of reads was counted. A total of 43,038 unigenes were assembled, with an average length of 1157 bp, an N50 value of 1685 bp, and a range of lengths from 301 to 13,934 bp. The Q30 score for each sample ranged between 94.13% and 94.72%, with an average GC content of 46.78%.

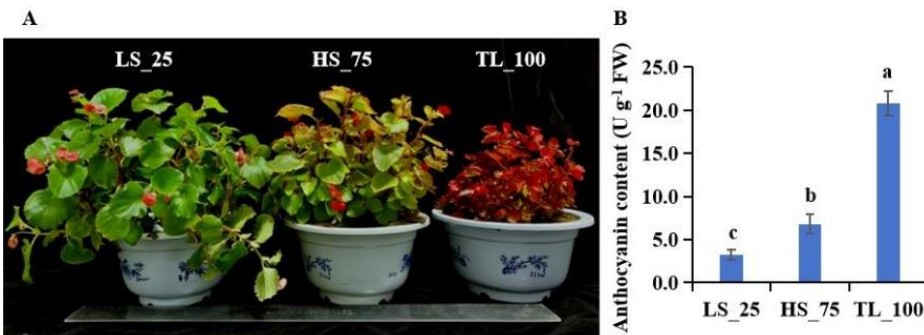

**Figure 1.** Phenotypes (**A**) and the relative content of anthocyanin (**B**) in the leaves of *B. semperflorens* under different light conditions: TL_100 (9500–9600 lx), HS_75 (6800–7000 lx), and LS_25 (4300–4500 lx). Values given in the form of mean $\pm$ SE ($n$ = 3). Significant differences were analyzed using Duncan's multiple range test; bars with different letters (a, b, c) are significantly different from each other ($p < 0.05$).

**Table 1.** Summary of sequencing data quality preprocessing results.

| Sample | Raw_Reads | Raw_Bases | Clean_Reads | Clean_Bases | Valid_Bases | Q30 | GC |
|---|---|---|---|---|---|---|---|
| SJHT01 | 49,777,226 | 7,466,583,900 | 48,004,270 | 6,990,744,218 | 93.63% | 94.40% | 46.57% |
| SJHT02 | 49,522,814 | 7,428,422,100 | 47,590,320 | 6,923,035,939 | 93.20% | 94.13% | 46.62% |
| SJHT03 | 49,863,826 | 7,479,573,900 | 47,936,466 | 6,967,194,631 | 93.15% | 94.15% | 46.63% |
| SJHT04 | 49,828,304 | 7,474,245,600 | 48,213,482 | 7,020,997,231 | 93.94% | 94.72% | 46.58% |
| SJHT05 | 49,100,084 | 7,365,012,600 | 47,345,236 | 6,871,630,987 | 93.30% | 94.43% | 46.35% |
| SJHT06 | 49,157,100 | 7,373,565,000 | 47,370,798 | 6,876,938,189 | 93.26% | 94.35% | 46.21% |
| SJHT07 | 49,563,616 | 7,434,542,400 | 47,804,554 | 6,944,073,974 | 93.40% | 94.45% | 47.48% |
| SJHT08 | 49,606,990 | 7,441,048,500 | 47,810,836 | 6,952,011,668 | 93.43% | 94.40% | 47.44% |
| SJHT09 | 49,281,318 | 7,392,197,700 | 47,370,676 | 6,890,183,691 | 93.21% | 94.24% | 47.13% |

For mapping clean reads to unigenes, Bowtie2 software (version 2.3.3.1) was utilized, showing a total read mapping rate of 100%, with over 80% of reads mapped and more than 60% uniquely mapped (Table 2) [25]. Pearson's correlation coefficient among biological replicates in each light condition indicated a high transcript abundance correlation (Figure S1). This was further supported by principal component analysis (PCA) results (Figure S2).

**Table 2.** Statistics of comparison results between reads and unigenes.

| Term/Sample | SJHT01 | SJHT02 | SJHT03 | SJHT04 | SJHT05 | SJHT06 | SJHT07 | SJHT08 | SJHT09 |
|---|---|---|---|---|---|---|---|---|---|
| Total reads | 48,004,270 (100.00%) | 47,590,320 (100.00%) | 47,936,466 (100.00%) | 48,213,482 (100.00%) | 47,345,236 (100.00%) | 47,370,798 (100.00%) | 47,804,554 (100.00%) | 47,810,836 (100.00%) | 47,370,676 (100.00%) |
| Total mapped reads | 41,984,877 (87.46%) | 41,550,401 (87.31%) | 41,907,962 (87.42%) | 41,832,385 (86.76%) | 41,286,182 (87.20%) | 41,353,105 (87.30%) | 41,068,040 (85.91%) | 41,291,670 (86.36%) | 40,733,631 (85.99%) |
| Multiple mapped | 11,293,504 (23.53%) | 10,918,104 (22.94%) | 11,014,508 (22.98%) | 10,837,352 (22.48%) | 10,429,495 (22.03%) | 10,401,779 (21.96%) | 10,774,523 (22.54%) | 10,746,716 (22.48%) | 10,275,731 (21.69%) |
| Uniquely mapped | 30,691,373 (63.93%) | 30,632,297 (64.37%) | 30,893,454 (64.45%) | 30,995,033 (64.29%) | 30,856,687 (65.17%) | 30,951,326 (65.34%) | 30,293,517 (63.37%) | 30,544,954 (63.89%) | 30,457,900 (64.30%) |
| Reads mapped in proper pairs | 38,922,888 (81.08%) | 38,559,570 (81.02%) | 38,869,636 (81.09%) | 38,862,724 (80.61%) | 38,360,006 (81.02%) | 38,436,858 (81.14%) | 37,948,358 (79.38%) | 38,199,484 (79.90%) | 37,668,310 (79.52%) |

### 3.2. Transcriptome Functional Annotation

The unigenes were compared with NR, KOG, GO, Swiss-Prot, eggNOG, and the KEGG database using diamond software (version 0.4.7) and the Pfam database via HMMER software (version 3.0) [23,24]. All of the unigenes were aligned to the NR, Swiss-Prot,

KEGG, KOG, eggNOG, GO, and Pfam protein databases. And the results showed that 31,197, 24,591, 12,096, 18,309, 30,226, 22,489, and 41 unigenes were identified in the NR, Swiss-Prot, KEGG, KOG, eggNOG, GO, and Pfam protein databases, respectively (Table 3). Most unigenes were annotated in NR protein databases, reaching 72.49%.

**Table 3.** Annotation of the unigenes in seven databases.

| Database | Annotated Number | Percentage (%) |
| --- | --- | --- |
| NR | 31,197 | 72.49 |
| Swiss-Prot | 24,591 | 57.14 |
| KEGG | 12,096 | 28.11 |
| KOG | 18,309 | 42.54 |
| eggNOG | 30,226 | 70.23 |
| GO | 22,489 | 52.25 |
| Pfam | 41 | 0.10 |

*3.3. Differentially Expressed Gene Analysis*

Differential gene expression was assessed using DESeq software (version 1.18.0) [32]. A total of 5411 DEGs were identified between HS_75 and TL_100, with 3078 up-regulated and 2333 down-regulated (Figure 2A; Table S2). In the LS_25 vs. TL_100 comparison, 4701 DEGs were identified, with 2648 up-regulated and 2053 down-regulated (Figure 2A; Table S3). Between LS_25 and HS_75, there were 6558 DEGs, with 3032 up-regulated and 3526 down-regulated (Figure 2A; Table S4). Additionally, 416 DEGs were commonly expressed across all three comparisons (Figure 2B; Table S5).

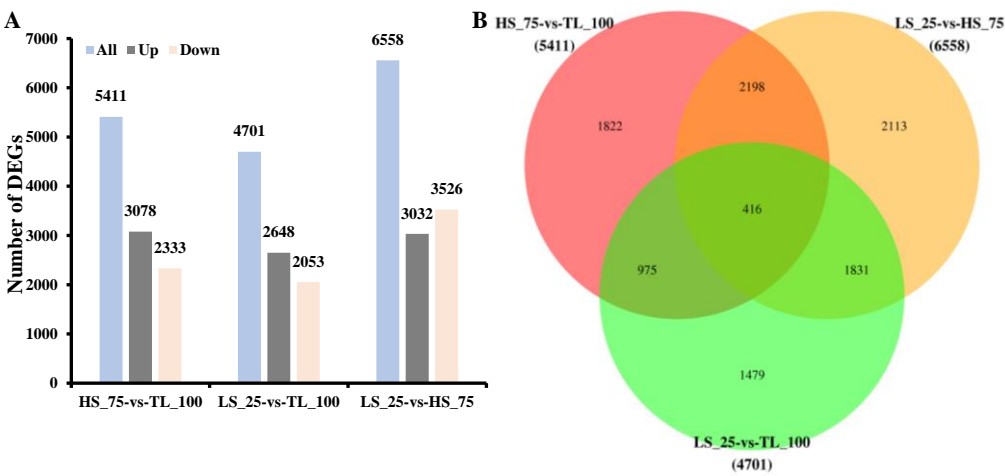

**Figure 2.** Number of up- or down-regulated genes in the three comparisons (**A**). The unigene venn diagram was expressed differently in each group (**B**).

3.3.1. GO Enrichment Analysis of DEGs

GO enrichment analysis was performed on the DEGs and results were integrated with those of GO annotations. The number of differential mRNAs produced in each GO term was counted and hypergeometric distribution tests were conducted to determine the significance of enrichment (Figure 3). The identified biological processes included flavonoid biosynthesis, pigment biosynthesis, and the regulation of flavonoid biosynthetic processes. Molecular functions related to enzymes involved in anthocyanin biosynthesis were also identified (Figure 3).

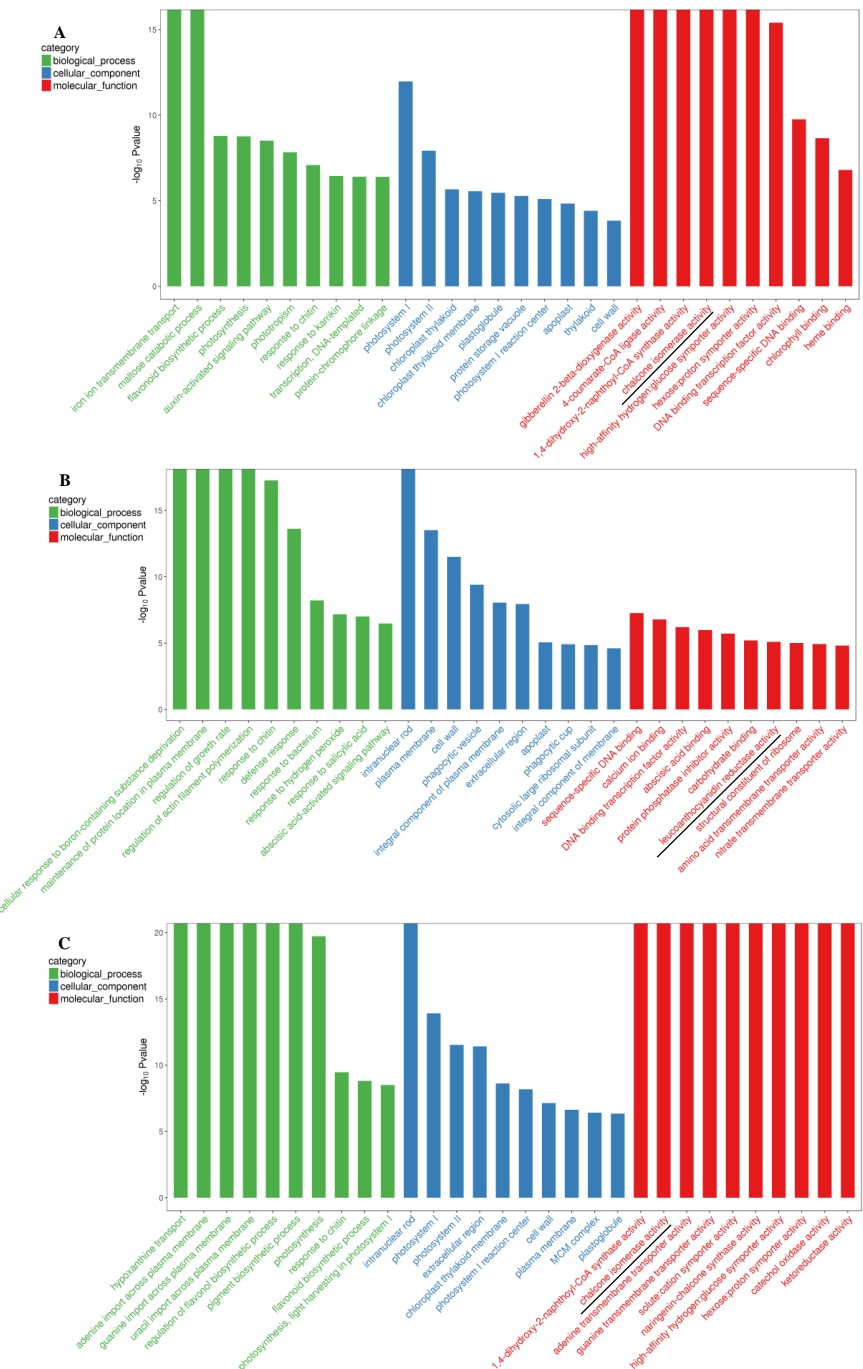

**Figure 3.** GO enrichment of top 30 terms of DEGs that were detected in the three comparisons of HS_75–vs–TL_100 (**A**), LS_25–vs–TL_100 (**B**), and LS_25–vs–HS_75 (**C**) in leaves. The blank lines indicate some anthocyanin biosynthesis-related enzymes. The horizontal coordinate shows the GO term and the vertical coordinate indicates the $-\log_{10} p$ value.

### 3.3.2. KEGG Enrichment Analysis of DEGs

KEGG pathway analysis was carried out to understand the pathways associated with the DEGs [33]. The top 20 enriched pathways are represented in a bubble diagram (Figure 4). Flavonoid metabolism, along with starch and sucrose metabolism and plant hormone signal transduction pathways, was found to be influenced by the light environment (Figure 4). The above results indicate that high-light stress can induce anthocyanin biosynthesis.

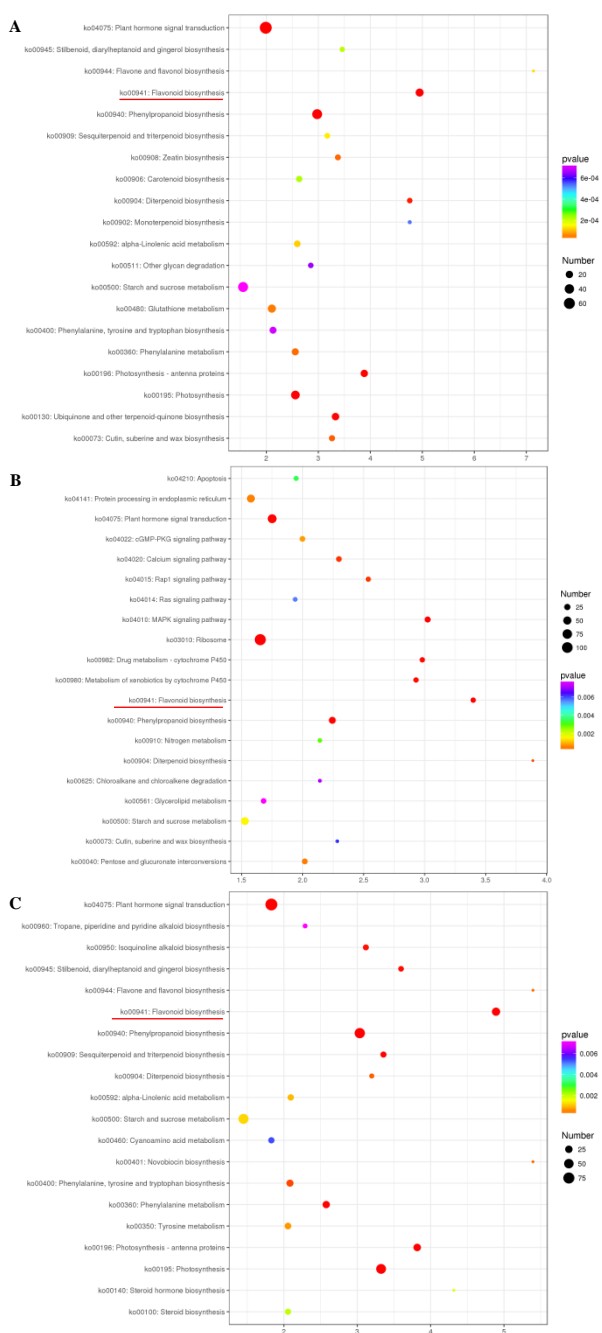

**Figure 4.** KEGG classification of DEGs that were detected in the three comparisons of HS_75–vs–TL_100 (**A**), LS_25–vs–TL_100 (**B**), and LS_25–vs–HS_75 (**C**). Red line shows the flavonoid biosynthesis pathway item. The *x*-axis is the enrichment score. The larger the bubble, the more the number of DEGs. The color of the bubble changes from purple to blue to green to red. The smaller the *p*-value of the bubble, the greater the significance.

### 3.4. Analysis of DEGs Involved in Flavonoid Biosynthesis

Within the study, several genes involved in phenylpropanoid biosynthesis, which leads to anthocyanin production, were identified. DEGs, including *CHS*, *CHI*, *F3H*, *F3′5′H*, *F3′H*, *DFR*, and *ANS*, were present in different conditions, whereas *3GT* (*UDP-glucose: flavonoid 3-o-glucosyltransferase*) was not detected. The expression of the *F3′H* gene was notably up-regulated with increased light intensity (Figure 5), suggesting its role in light-induced anthocyanin biosynthesis.

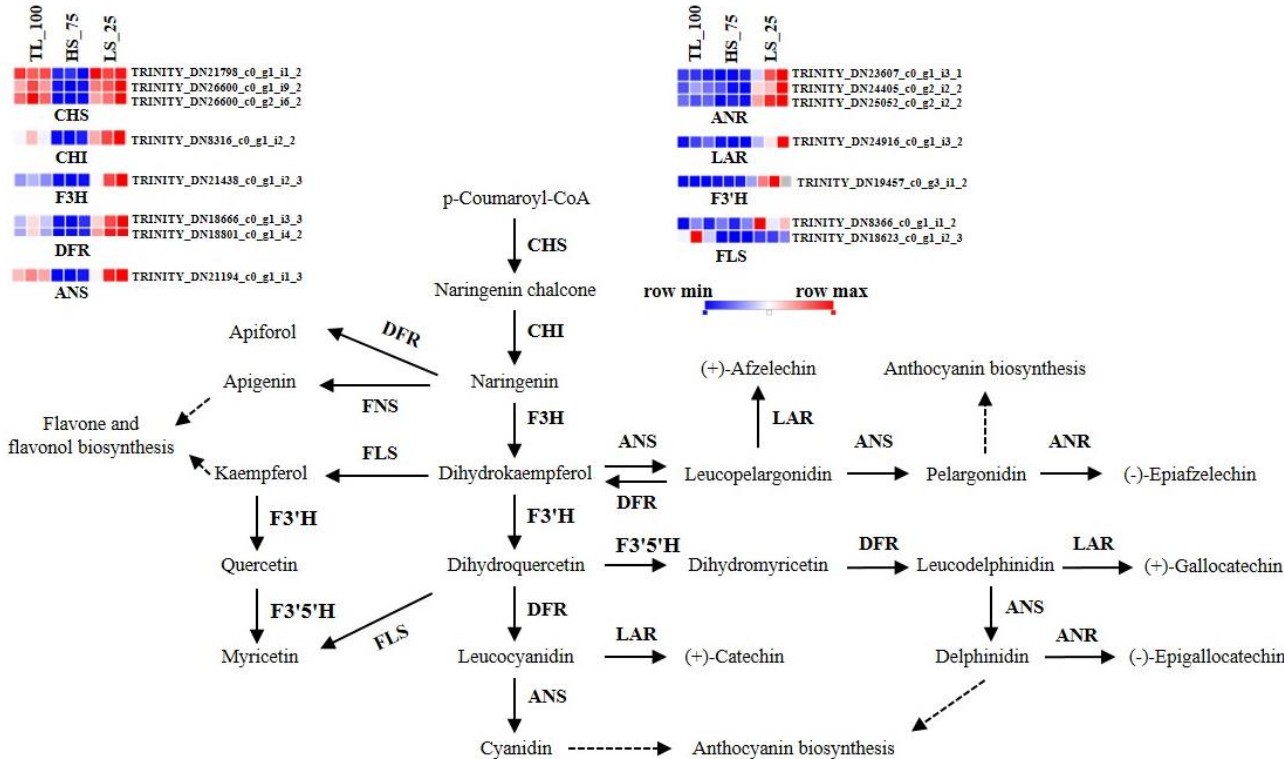

**Figure 5.** Schematic of physiological roles and the expression of DEGs regulated by different light conditions in flavonoid biosynthesis in *B. semperflorens*. *CHS*, chalcone synthase; *CHI*, chalcone isomerase; *DFR*, dihydroflavonol 4–reductase; *F3H*, flavanone 3–hydroxylase; *F3′H*, flavanone–3–hydroxylase; *F3′5′H*, flavonoid F3′5′H–hydroxylase; *FNS*, flavone synthase; *LAR*, leucoanthocyanidin reductase; *ANR*, anthocyanidin reductase; *FLS*, flavonol synthase.

### 3.5. Transcription Factor Database Annotation

Transcription factors are proteins that bind to specific DNA sequences, which play a key role in gene expression regulation by controlling the transcription of genetic information from DNA to RNA. PlantTFDB (https://opendata.pku.edu.cn/dataset.xhtml?persistentId=doi:10.18170/DVN/GHICUF, accessed on 10 January 2024) is a database of plant transcription factors which contains sequences from 165 plants and 58 families of plant transcription factors, and the top 10 species distribution is shown in Figure S3. The unigene sequence was compared to the transcription factor database by blastx (E-value $< 1 \times 10^{-5}$). A total of 16,363 unigenes were annotated in the transcription factor database, which were distributed in 58 families. The TF comparison results are shown in Figure 6, and the MYB and MYB-related, bHLH (basic Helix–Loop–Helix), WRKY (WRKYGQK), and ERF (Ethylene-Responsive Factor) transcription factors showed a higher number.

Then, the expression levels of some key candidate DEGs related with anthocyanin biosynthesis were further examined under light conditions (Table S6). The expression of the *CRF6* (*Cytokinin Response Factor 6*), *ERF2*, *F3′H*, *MYB102*, *WRKY40*, *WRKY76*, *WRKY53*, and *WRKY70* genes was up-regulated in HS_75-vs-TL_100, LS_25-vs-TL_100, and LS_25-vs-HS_75; however, the expression of *SWEET1* (*Sugars Will Eventually be Exported Transporters*), *ATHB12* (*Arabidopsis thaliana homeobox genes*), *HSP20* (*Heat Shock Protein 20*), and *HSP-like* was down-regulated (Figure 6). In short, transcription factors may also be involved in regulating the anthocyanin biosynthesis of *B. semperflorens* under light conditions.

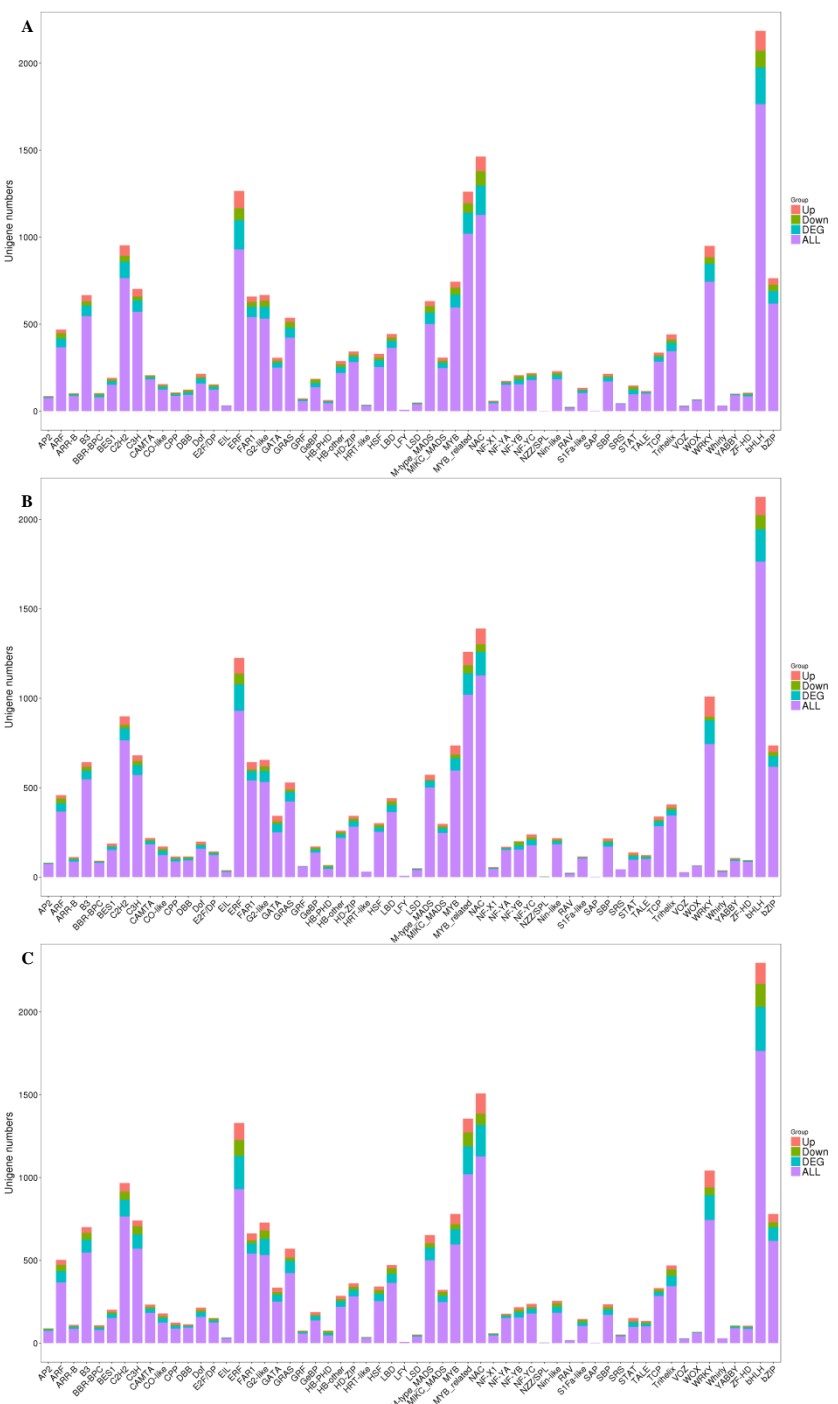

**Figure 6.** The distribution of the transcription factor family that were detected in the three comparisons of HS_75–vs–TL_100 (**A**), LS_25–vs–TL_100 (**B**), and LS_25–vs–HS_75 (**C**). Purple represents all unigenes, blue represents differentially expressed unigenes, green represents down-regulation of unigenes, and red represents up-regulation of unigenes. The abscissa is the transcription factor family and the ordinate is the number of unigenes.

### 3.6. Transcriptome Data Were Verified via qRT-PCR

In order to verify the validity of the data of transcriptomes, we selected 12 genes (*ERF2, MYB102, F3′H, WRKY40, WRKY76, WRKY53, WRKY70, SWEET1, CHI, F3H, ATHB12,* and *HSP-like*) from the DEGs for qRT-PCR analysis of their expression under different light conditions (Figure 7). These results are consistent with the data of transcriptomes. Meanwhile, we also found that the *ERF2, MYB102, F3′H, WRKY40, WRKY76, WRKY53,* and

*WRKY70* genes were up-regulated with increases in light intensity (Figure 7); in contrast, the *SWEET1*, *ATHB12*, and *HSP-like* genes decreased with increases in light intensity (Figure 7).

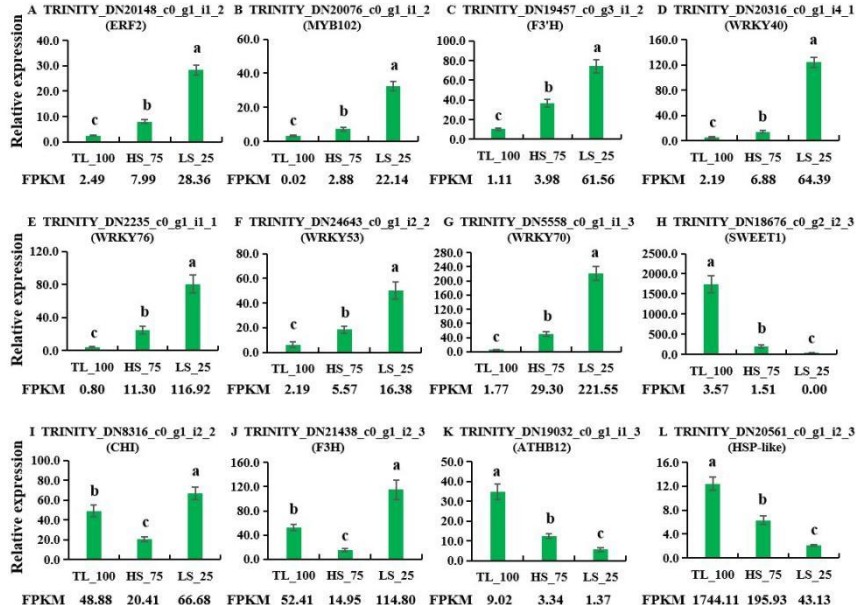

**Figure 7.** qRT-PCR validation of the transcriptome data. The relative transcript levels of these twelve genes were shown in panel (**A**) (*ERF2*), (**B**) (*MYB102*), (**C**) (*F3'H*), (**D**) (*WRKY40*), (**E**) (*WRKY76*), (**F**) (*WRKY53*), (**G**) (*WRKY70*), (**H**) (*SWEET1*), (**I**) (*CHI*), (**J**) (*F3H*), (**K**) (*ATHB12*), and (**L**) (*HSP-like*). Values given in the form of mean $\pm$ SE ($n$ = 3). Significant differences were analyzed using Duncan's multiple range test; bars with different letters are significantly different from each other ($p < 0.05$). The sequences of primers are shown in Table S7.

## 4. Discussion

### 4.1. Anthocyanin Was Induced by Light Conditions

Anthocyanins, as significant flavonoid compounds, play a critical role in plant defense mechanisms and adaptability to various environmental stresses [18,34]. The biosynthesis of anthocyanins in *B. semperflorens* 'Super Olympia' seedlings is induced by low temperatures, alongside the synthesis of lignin [18]. Similarly, anthocyanin content in grapes increases upon infection with *Colletotrichum gloeosporioides* [34]. Previous studies, including ours, have shown that shading can alter both the morphological and physiological characteristics of plants [5,20]. High light induces an accumulation of superoxide anions ($O_2^-$) and malondialdehyde (MDA); meanwhile, the relative contents of anthocyanin, soluble sugar, starch, and superoxide dismutase (SOD) activity also increased [20]. Fortunately, under high-light conditions, the color of *B. semperflorens* leaves turned to red, which has high ornamental value (Figure 1A).

Generally, the structural genes, like *CHI*, *CHS*, *DFR*, and *ANS*, within the flavonoid metabolic pathway significantly influence the flavonoid content [35,36]. High light is known to induce anthocyanin biosynthesis by up-regulating these structural genes, such as *PAL*, *CHS*, *F3H*, and *ANS* genes [18]. And our study reveals a novel aspect, where *F3'H* gene expression was markedly up-regulated with increased light intensity (Table S6 and Figure 7). So far, the *F3'H* gene has been reported to positively regulate anthocyanin biosynthesis. In the present study, the anthocyanin content was 11.54 $\pm$ 1.99 U·g$^{-1}$·FW, 45.42 $\pm$ 3.43 U·g$^{-1}$·FW, and 134.60 $\pm$ 6.4 U·g$^{-1}$·FW in LS_25, HS_75, and TL_100, respectively (Figure 1B). Therefore, the *F3'H* gene is responsible for anthocyanin synthesis and then further resistance to high-light stress. And, in *B. semperflorens*, it was also reported that exogenous sucrose could induce anthocyanin accumulation, which is related to the up-regulation of PAL, CHI, DFR, etc. [37]. Hence, the function of the *F3'H* gene must be further explored.

*4.2. High Light Affects the Expression of Transcription Factors*

It is well known that transcription factors also play key roles in modulating anthocyanin biosynthesis under stress conditions [38,39]. For example, in *Lycoris radiata*, Lr-WRKY3/27 regulates anthocyanin synthesis under drought stress [39]. In *Malus* crabapple, the McWRKY71 transcription factor is induced by ozone stress, influencing anthocyanin and proanthocyanidin synthesis [40]. The bHLH137 transcription factor is involved in the positive regulation of proanthocyanidins and anthocyanins in grapevines, enhancing resistance to *C. gloeosporioides* [34]. Additionally, the postharvest temperature and light treatments in 'Akihime' plum (*Prunus salicina* Lindl.) induce anthocyanin accumulation via the transcription factor PsMYB10.1 [41]. In the Asiatic hybrid lily (*Lilium* spp.), LhWRKY44 is implicated in light- and drought-induced anthocyanin synthesis [12]. Furthermore, a model of light-regulated anthocyanin biosynthesis in rose petals included a network of genes, such as *HY5* (*ELONGATED HYPOCOTYL 5*), *MYB114a*, *bHLH3*, *CHS*, and *F3′H* [42].

In the present study, we observed that high-light exposure not only induces anthocyanin biosynthesis-related genes but also affects the expression levels of various stress-responsive genes (Table 4), like *WRKY*, *HSP-like*, *ERF* genes, etc. Meanwhile, in order to resist the high-light environment, we found that the leaf area decreased and curled, the flower number decreased, the plant height reduced, and so on (Figure 1A). These results imply that the pant growth was also affected by these transcription factors, such as *AP1* (*APETALA1*), *ERF2*, *SWEET1*, etc. And these transcription factors may have functions in responding to high-light environments. However, the lack of transgenic systems makes it difficult to clearly explain their function.

**Table 4.** Some key transcription factors that respond to high-light conditions.

| Gene_Id | HS_75–vs–TL_100 | LS_25–vs–HS_75 | LS_25–vs–TL_100 | Annotation |
|---|---|---|---|---|
| TRINITY_DN17817_c0_g1_i3_2 | Up | Up | Up | *WRKY70* |
| TRINITY_DN18363_c1_g1_i1_2 | Up | Up | Up | *CRF6* |
| TRINITY_DN19191_c0_g1_i3_2 | Up | Up | Up | *WRKY-like* |
| TRINITY_DN19457_c0_g3_i1_2 | Up | Up | Up | *F3′H* |
| TRINITY_DN20076_c0_g1_i1_2 | Up | Up | Up | *MYB102* |
| TRINITY_DN20148_c0_g1_i1_2 | Up | Up | Up | *ERF2* |
| TRINITY_DN20316_c0_g1_i4_1 | Up | Up | Up | *WRKY40* |
| TRINITY_DN21459_c0_g1_i15_2 | Up | Up | Up | *WRKY40* |
| TRINITY_DN2235_c0_g1_i1_1 | Up | Up | Up | *WRKY76* |
| TRINITY_DN24643_c0_g1_i2_2 | Up | Up | Up | *WRKY53* |
| TRINITY_DN5558_c0_g1_i1_3 | Up | Up | Up | *WRKY70* |
| TRINITY_DN14802_c0_g1_i1_3 | Down | Down | Down | *AP1* |
| TRINITY_DN18676_c0_g2_i2_3 | Down | Down | Down | *SWEET1* |
| TRINITY_DN19032_c0_g1_i1_3 | Down | Down | Down | *ATHB12* |
| TRINITY_DN20561_c0_g1_i2_3 | Down | Down | Down | *HSP20-like* |
| TRINITY_DN24640_c0_g3_i1_3 | Down | Down | Down | *HSP-like* |

Significant differences were determined with a *p*-value < 0.05 and | log2 fold change| > 1.

## 5. Conclusions

In summary, our research demonstrates that high-light exposure triggers anthocyanin accumulation in the leaves of adult *B. semperflorens* plants. The identification of DEGs under high-light conditions revealed that anthocyanin accumulation is a strategic response to high-light stress. Importantly, these DEGs not only include structural genes but also transcription factors (Table 4; Figure 8). High light promoted the expression of *ERF2*, *MYB102*, *F3′H*, *WRKYs*, etc., but also inhibited the expression of *SWEET1*, *ATHB12*, and *HSP-like* genes. This suggests that these genes may be involved in both anthocyanin biosynthesis and the response to high-light stress environments.

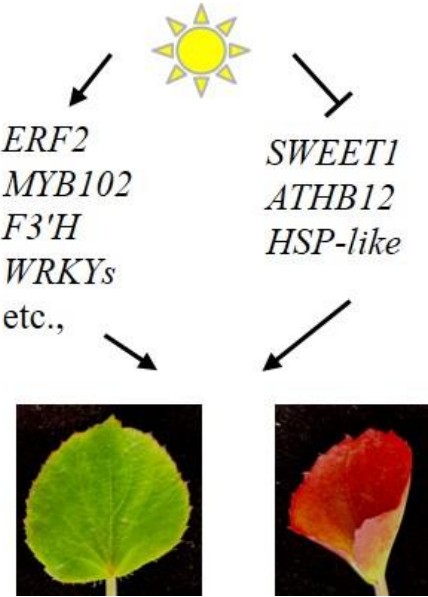

**Figure 8.** Working model of light-regulated DEGs in *B. semperflorens*.

**Supplementary Materials:** The following supporting information can be downloaded at https://www.mdpi.com/article/10.3390/horticulturae10010096/s1: Table S1: The results of transcriptome splicing; Figure S1: Pearson's correlation between nine samples; Figure S2: Principal component analysis; Table S2: DEGs in HS_75–vs–TL_100; Table S3: DEGs in LS_25–vs–TL_100; Table S4: DEGs in HS_75–vs–TL_100; Table S5: DEGs in the three comparisons; Figure S3: Transcription factor alignment results; Table S6: The differentially expressed genes; Table S7: The sequences of primer used in this study.

**Author Contributions:** Conceptualization, A.L. and Y.Z.; methodology, Z.Z. and W.L.; software, S.Y.; validation, Z.Z. and W.L.; formal analysis, K.Z.; investigation, K.Z.; resources, A.L. and W.L.; data curation, A.L.; writing—original draft preparation, K.Z. and A.L.; writing—review and editing, K.Z. and A.L.; visualization, K.Z. and Z.Z.; supervision, A.L.; project administration, Y.Z. and A.L.; funding acquisition, W.L. and Y.Z. All authors have read and agreed to the published version of the manuscript.

**Funding:** This study was financially supported by grants from the Key Projects of Natural Science Research in colleges and universities of Anhui (2022AH051621, 2022AH051622), the Science and Technology Plan Project of the Anhui Housing Construction Department (2022-YF038), the Key Research Project of Chuzhou (2022ZN015), the Introduction of Talent Projects of Anhui Science and Technology University (JZYJ202201), and the Natural Science Foundation of Zhejiang Province (Q22C158467).

**Data Availability Statement:** All data included in this study are available upon request through contact with the corresponding author. The data are not publicly available due to the use in other unpublished articles.

**Conflicts of Interest:** The authors declare no conflicts of interest.

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
