# Peer review of "Comparative Transcriptome Analysis Reveals Changes in Gene Expression Associated with Anthocyanin Metabolism in Begonia semperflorens under Light Conditions"

_horticulturae, doi:10.3390/horticulturae10010096_

Round 1
Reviewer 1 Report
Comments and Suggestions for Authors
I reviewed the manuscript entitled "Comparative transcriptome analysis reveals changes of gene expression associated with anthocyanins metabolism in Begonia semperflorens under light conditions". The text is relatively good, but the authors need to improve the text before publishing.

Comments on the Quality of English LanguageImprove spelling.
Author Response
I have revised it again. Please see the attachment.

Reviewer 2 Report
Comments and Suggestions for Authors
Introduction
Line 71 - I would suggest the authors expand this part of the Introduction. Transcriptomics is only briefly mentioned, although it is an important part of this study.
Line 75 - I think that the Introduction (and the Discussion) lack a commentary on mechanisms of light-induced stress in plants, specifically begonias. What is already known about light-induced stress? What are physiological changes and how are they connected to gene expression? What is the same and what is different between "normal" light conditions and stress-inducing light conditions? I think this is not sufficiently explored in the Introduction (and Discussion).
M & M, Results
Line 114 - in general, aim to use passive forms of verbs instead of active forms ("1 microgram of RNA was used" instead of "we used 1 microgram of RNA").
Line 222 - same comment.
Line 239 - were these genes chosen randomly or was there some criterion for their selection? The authors should clarify this.
Discussion - as a general remark, it is not customary to reference figures and tables from your results in the discussion. You should focus on discussing the results that you already presented graphically and textually. Also, consider using Figure 8 as a graphical abstract for your manuscript. Furthermore, you need to emphasize the limitations of your current study and propose solutions for future research.
Line 259 - this part should be expanded, too. Do you have a hypothetical explanation as to why the F3'H gene is that much up-regulated in your experiments? This is the gene for flavanone 3'-hydroxylase, right? This should be mentioned here, with a short explanation of its role in flavonoid biosynthesis. How does the increase in light lead to an increase in the expression of this gene? If this is the novelty of your research, you should explain it more and highlight it.
Line 273 - I think a discussion on the mechanisms of light-induced stress is missing in this section. How are normal physiological processes affected under stress and how does the plant counteract that? How does the light stress affect the gene expression? How do your results compare to what is already known about this and what is new in your research? Also, the authors used a diagram in Figure 5 in the Results to highlight the role of major genes responsible for anthocyanin biosynthesis in response to light. In my opinion, a theoretical discussion of these pathways is lacking in the Discussion.
Conclusions - I am not sure if such a comparison between normal light conditions and stress-inducing light conditions can be made based on the results of your research. Try to re-phrase this, perhaps by making a distinction between genes and their expression under "normal" and "abnormal" conditions.
Comments on the Quality of English LanguageI suggest only minor English language editing.
Author Response
Line 71 - I would suggest the authors expand this part of the Introduction. Transcriptomics is only briefly mentioned, although it is an important part of this study.
Response: Thank you for pointing this out. I agree with this comment. And I have expanded this part of the Introduction in the text (line 72-78).
Line 75 - I think that the Introduction (and the Discussion) lack a commentary on mechanisms of light-induced stress in plants, specifically begonias. What is already known about light-induced stress? What are physiological changes and how are they connected to gene expression? What is the same and what is different between "normal" light conditions and stress-inducing light conditions? I think this is not sufficiently explored in the Introduction (and Discussion).
Response: Thank you for your valuable advice. These questions are very great for us. An we describe more information in the text. (showed in line 275-278, line 83-85, line 279-283, 289-297) M & M, Results
Line 114 - in general, aim to use passive forms of verbs instead of active forms ("1 microgram of RNA was used" instead of "we used 1 microgram of RNA").
Response: Thank you for your valuable advice. I agree that, and I changed it (line 126).
Line 222 - same comment.
Response: I quite agree with your suggestion. And I revised it (line 244).
Line 239 - were these genes chosen randomly or was there some criterion for their selection? The authors should clarify this.
Response: Thank you for pointing this out. Because I think these genes are important, and I will explore their function next time.
Discussion - as a general remark, it is not customary to reference figures and tables from your results in the discussion. You should focus on discussing the results that you already presented graphically and textually. Also, consider using Figure 8 as a graphical abstract for your manuscript. Furthermore, you need to emphasize the limitations of your current study and propose solutions for future research.
Response: Thank you for your valuable advice. I have modified them in corresponding places in the text (line 279-322).
Line 259 - this part should be expanded, too. Do you have a hypothetical explanation as to why the F3'H gene is that much up-regulated in your experiments? This is the gene for flavanone 3'-hydroxylase, right? This should be mentioned here, with a short explanation of its role in flavonoid biosynthesis. How does the increase in light lead to an increase in the expression of this gene? If this is the novelty of your research, you should explain it more and highlight it.
Response: Thank you for pointing this out. And I have modified them in the text (line 289-297).
Line 273 - I think a discussion on the mechanisms of light-induced stress is missing in this section. How are normal physiological processes affected under stress and how does the plant counteract that? How does the light stress affect the gene expression? How do your results compare to what is already known about this and what is new in your research? Also, the authors used a diagram in Figure 5 in the Results to highlight the role of major genes responsible for anthocyanin biosynthesis in response to light. In my opinion, a theoretical discussion of these pathways is lacking in the Discussion.
Response: Thank you for your valuable advice. And I have modified them in the text (line 312-320).
Conclusions - I am not sure if such a comparison between normal light conditions and stress-inducing light conditions can be made based on the results of your research. Try to re-phrase this, perhaps by making a distinction between genes and their expression under "normal" and "abnormal" conditions.
Response: Thank you for your valuable advice. Table 4 was added. And I re-phrase the Conclusions (line 324-332).
Comments on the Quality of English Language I suggest only minor English language editing.
Response: Thank you for pointing this out. And I had do minor English language editing. And the revised text in the attachment.

Round 2
Reviewer 2 Report
Comments and Suggestions for Authors
I see that the Authors made many corrections that were suggested to them. I think that the manuscript is improved. However, there are language issues that have to be addressed.
Comments on the Quality of English LanguageI strongly recommend that a native-level English speaker revise the entire manuscript. Issues with grammar, spelling, and style should be corrected before publication.
Author Response
I have revised it and conducted english editing.
